# Nitric Oxide Interaction with the Eye

**Nir Erdinest** [1] , **Naomi London** [2,*] , **Haim Ovadia** [3] and **Nadav Levinger** [1,4]

1   Department of Ophthalmology, Hadassah-Hebrew University Medical Center, Jerusalem 91120, Israel; nir.erdinest@mail.huji.ac.il (N.E.); nadavlevinger@gmail.com (N.L.)
2   Private Pracitce, Jerusalem 94228, Israel
3   Agnes Ginges, Center for Human Neurogenetics, Department of Neurology, Hadassah-Hebrew University Medical Center, Jerusalem 91120, Israel; ovadiafam@gmail.com
4   Enaim Refractive Surgery Center, Jerusalem 9438307, Israel
*   Correspondence: imnl4u@gmail.com

**Abstract:** Nitric oxide (NO) is acknowledged as a vital intercellular messenger in multiple systems in the body. Medicine has focused on its functions and therapeutic applications for decades, especially in cardiovascular and nervous systems, and its role in immunological responses. This review was composed to demonstrate the prevalence of NO in components of the ocular system, including corneal cells and multiple cells in the retina. It discussed NO's assistance during the immune, inflammation and wound-healing processes. NO is identified as a vascular endothelial relaxant that can alter the choroidal blood flow and prompt or suppress vascular changes in age-related macular degeneration and diabetes, as well as the blood supply to the optic nerve, possibly influencing the progression of glaucoma. It will provide a deeper understanding of the role of NO in ocular homeostasis, the delicate balance between overproduction or underproduction and the effect on the processes from aqueous outflow and subsequent intraocular pressure to axial elongation and the development of myopia. This review also recognized the research and investigation of therapies being developed to target the NO complex and treat various ocular diseases.

**Keywords:** nitric oxide; myopia; glaucoma

## 1. Introduction

Nitric oxide (NO) was first discovered in the 1770s by Joseph Priestly in England [1,2]. NO's medicinal benefit was essentially ignored until the early 1900s, as it was believed to be an air pollutant [2–4]. Then, medicine began using nitrates (e.g., nitroglycerin) for angina, and pharmacologists started to outline the physiologic responses of various tissues to these compounds, specifically in the cardiovascular field, where the improvement of angina pectoris and reversal of ischemia were reported. It was discovered to have a relaxing effect on respiratory and gastrointestinal smooth muscle tissues as well, and from there, clinicians moved to treat other smooth muscle tissues, such as reactive airway disease [5–8]. Since Furchgott, Ignarro and Murad were awarded the Nobel Prize in Physiology or Medicine in 1998 for their work "concerning nitric oxide as a signaling molecule in the cardiovascular system", much more research has been done to understand the critical roles of NO [2–4]. Knowledge of the tolerance and dosing of compounds as therapeutics has expanded, and NO is recognized as a potent vasodilator and endothelium-derived relaxing factor (EDRF) with the ability to impact multiple systems in the body [9–17].

NO is generated both inter- and intracellularly, its gaseous nature allowing it to diffuse through cell membranes. It is a free radical that plays a role in the vasodilatation of smooth muscle, neurotransmission and cytotoxicity [6].

There are three isoforms of nitric oxide synthase (NOS); each has a particular function. The endothelial (NOS-3) and neuronal (NOS-1) enzymes are calcium-dependent, which produce low levels of NO as a cell signaling molecule in resting cells [6,13,18]. Another isoform of the enzyme is the inducible calcium-independent isoform (NOS-2) that is

responsible for the release of NO during inflammation and is upregulated by a variety of extracellular stimuli, such as interleukin-1β (IL-1β), tumor necrosis factor-α (TNF-α) and LPS [6,13,18].

Inducible NO has an additional important role in the immune and inflammatory responses, contributing to the acute immune response via two distinct pathways [7,19]. The first pathway is direct, in which NO in the presence of $O_2$ produces another radical HNOO that has a toxic effect against infectious organisms as part of the innate immune system [7,19]. The second is indirect, in which NO is capable of inducing or regulating the function of immune cells as part of the specific immune process [20–22]. Previous studies related to NO's effect on the ocular surface suggested several roles for NO, such as cell damage during infection, the pathogenesis of endotoxin-induced uveitis, inhibiting neovascularization, producing corneal edema and inducing allergic reactions [14,17,23–26].

The generation of NO and ROS within cells may generate even more reactive radicals such as peroxynitrite. Peroxynitrite is capable of nitrating and oxidizing proteins, inferring a considerable impact on the integrity of cellular functions [27]. Peroxynitrite affects signaling pathways such as mitogen-activated protein kinase (MAPK)/Akt, while the nitration of tyrosine residues modulates the signaling processes relying on tyrosine phosphorylation and dephosphorylation; the oxidation of phosphotyrosine phosphatases may lead to an alteration in the tyrosine phosphorylation/dephosphorylation balance [27]. Peroxynitrite was demonstrated to activate the p38 and Jun N-terminal kinases (JNK) and extracellular signal-regulated kinases (ERK) 1/2 in a wide variety of cell types. Consequently, the expression of stress genes such as c-fos and heme oxygenase-1 is also induced [27,28]. ERK activation by peroxynitrite in human neutrophils leads to the expression of CD11b/CD18 and to the enhanced adhesion of peroxynitrite-treated leukocytes to lipopolysaccharide-treated endothelial cells [27,28].

NO may affect the activity of enzymatic antioxidants via the peroxynitrite-mediated mechanism. For example, the reduction of superoxide dismutase (SOD) activity can reduce the removal of superoxide anions and produce $H_2O_2$. The same peroxynitrite pathway can alter the catalase activity, thus lowering the cell's capacity to remove $H_2O_2$, perhaps prolonging the ROS-mediated signaling [27,28].

The antioxidant capacity of the cell is due to the presence of low molecular weight molecules such as glutathione. NO and glutathione can react together to produce S-nitrosoglutathione (GSNO). The reaction will also remove NO during further signaling. GSNO can act also as a donor of NO, and it has been suggested that GSNO can mediate some NO effects [27,28].

This review discusses the primary NO affiliations in the eye regarding homeostasis and its role in the prevention and causation of common ocular diseases and inflammation such as corneal wounds, glaucoma, age-related macular degeneration (AMD) and myopia. Included as well are the NO complex targeted treatments in development published in the literature as alternatives to the current available options.

## 2. NO and the Eye

NO is produced pre- and post-synaptically in the nervous system, and while beyond the scope of this review, it behooves mentioning that NO physiologically influences the visual system posterior to the eye at the lateral geniculate nucleus (LGN) and in the primary visual cortex [29].

Within the ocular globe, NO plays an important role in both the anterior and posterior segments.

The underproduction of NO results in various eye diseases. On the other hand, immunological NOS (iNOS) is inducible only in pathological conditions [11,30,31]. Once induced, iNOS will produce large amounts of NO for long periods of time, so that NO is converted into $NO_2$, nitrite, peroxynitrite and free radicals, which induce pathophysiological actions to treat or even prevent eye disease onset; inhibitors of iNOS activity and/or iNOS induction could be tried [30,31].

In addition, NO affects eye development, as is evident in the *Drosophila* model. Research found that manipulations of endogenous or transgenic NOS activity during imaginal disc development can either enhance or suppress eye development [32]. Furthermore, the increased production of NO acts as an antiproliferative signal, whereas the inhibition of NOS activity promotes additional rounds of cell division in eye development [32].

In summary, NO can either have a protective or toxic effect, depending on the situation and concentration.

### 3. NO in Ocular Surface Cells

The main sources of NO in ocular surface tissue are the corneal epithelium, fibroblast, endothelium and inflammatory cells [33,34].

Kim et al. [33] induced ocular inflammation in rabbits in vivo and demonstrated that stromal fibroblasts and inflammatory cells are the primary sources of NO in ocular inflammation [35]. They also examined the NO level in tears. According to this study, if the concentration ratio of NO is 1.5–2.5-fold higher than the normal NO functional level (defined as 1.0), NO may play a defensive role. However, if the concentration ratio of NO is 3–10-fold higher, NO may induce tissue damage [35]. Targeted cells then activate guanylate cyclase and generate cGMP. This process is what leads to smooth muscle relaxation [36–38].

The ability of cells and tissues to regulate the production and accumulation of NO in large amounts (10 mM or greater) is, by NOS-2, a 130-kDa protein that is primarily regulated at the transcriptional level [12]. A study documented the production of NOS-2 mRNA expression and nitrite accumulation in cytokines and lipopolysaccharide (LPS)-treated human conjunctival fibroblasts (HCF) and corneal epithelial (HCE) cells cultured for four passages [39]. The elevated NOS-2 expression coincided in time with the accumulation of nitrites. The ability to induce NOS-2 in HCF required the presence of proinflammatory cytokines and LPS. When comparing the nitrite accumulation to the NOS-2 mRNA accumulation in HCF treated with the same cytokine combination, there was a mismatch between the significantly elevated NOS-2 expression compared to the relatively slight rise in nitrite levels in the medium culture [39]. Vodovotz et al. noted that mouse peritoneal macrophages incubated with IFN-$\gamma$ and LPS, though initially producing the NOS-2 protein and NO, eventually have a post-translational and nondegradative inactivation of the enzyme [10]. These cultured cells with inactivated enzymes are somewhat comparable with what was observed in HCF—the expression of high levels of NOS-2 mRNA compared to low NO production. This group also noted that the inactivation of NOS-2 required the presence of LPS [10].

The clinical data from studies on the systemic effects of orally administered n-3 PUFA, within an in vivo murine study, demonstrated significant systemic and local anti-inflammatory effects of n-3 polyunsaturated fatty acids (PUFAs) [40].

### 4. NO in the Retina

NO in retinal cells acts as a light response modulator by activating specific ion conductors in rods, cones and bipolar and ganglion cells [41]. It was observed in neurons and the pigment epithelium, indicating a role in the development or protection from infection in the ganglion cell layer and in the nuclear layers of the retina [42,43]. It has been suggested that amacrine cells may be the most prominent source for NO but has also been sourced from certain bipolar cells, which implies involvement in both in the physiological and pathological processes here as well [43,44]. Other studies have demonstrated that NO acts as a vascular endothelial relaxant involved in normal retinal blood flow control, as well as related to several ocular diseases associated with oxidative stress, including retinitis pigmentosa, diabetic retinopathy, glaucoma and AMD [42–44].

### 5. NO for the Treatment of Corneal Wound Healing

The wound-healing effects of NO are well-known. One study showed that a topical treatment with $NaNO_2$ (10 $\mu$M) enhanced the corneal epithelial recovery, as well as

decreased corneal opacity, in a murine corneal alkali burn created by modulating inflammatory cytokines [45].

## 6. NO and Intraocular Pressure (IOP)

NO has shown to be an IOP regulator. IOP is a balance between aqueous humor production and outflow through the uveoscleral network and the pressure-dependent trabecular meshwork pathway [36,46]. NO and its messenger cGMP directly act on trabecular meshwork cells to help increase the outflow. Studies have exhibited similarities between trabecular cells and smooth muscle cells, and both are very reactive to NO [47–50]. Stress or pressure trigger the production of NO, which then leads to the relaxation and dilation of the distal outflow tract, thus lowering the IOP. There are indications that NO is also produced in Schlemm's canal cells, indicating a direct regulatory role, as well as evidence that NO has a dilatory effect as well on myosin-containing cells on the outer wall of Schlemm's canal distal to the trabecular meshwork [47–50].

The documentation of an impaired NO regulatory system is compatible with the findings of decreased concentrations of NO and cGMP in the plasma and aqueous humor in glaucoma patients [51,52]. An outflow resistance and subsequent elevation in IOP has been exhibited in a number of studies by altering the NO system and lowering its production using various genetic modifications or pharmacological alterations [51].

## 7. NO for the Treatment of Glaucoma

Many existing therapies for IOP reduction are beneficial for a period of time, after which individuals require additional drugs to prevent the progression of nerve death [53]. One of the challenges to create therapies including NO is its short half-life in tissues of two seconds and only two milliseconds in blood [34,54]. A way to overcome this is to utilize known effective drugs for lowering IOP combined with NO-donor substances that are metabolized inside the eye and then release NO in the targeted tissue [46,49,55]. Some of the promising NO donor drugs being developed and thoroughly investigated are Latanoprostene bunod 0.024% (Bausch and Lomb) in a phase 3 trial, which is designed to increase the aqueous outflow both by uveoscleral and trabecular pathways, and are also being compared to other conventional glaucoma drugs, as well as various dosage efficacies [56–58]. A modified prostaglandin F2$\alpha$ analog bimatoprost carries a NO moiety and two other compounds, NCX 139, a NO-donor latanoprost amide, and NCX 125, a NO-donor latanoprost-free acid [56–60].

Mesoporous silica nanoparticles have a low toxicity and high drug-loading capacity. When filled with sodium nitroprusside, they have shown, in animal models, that they can produce more exogenous NO and sustain higher NO concentrations than the other available and potentially available drugs and, therefore, have the capacity to extend the duration of the reduction of IOP from 3 to 48 h with only 1/40 of the dose required for the same effect through other drug vehicles [60,61].

It has been suggested that primary open-angle glaucoma may be associated with structural abnormalities or irregularities of the blood vessels supplying the optic nerve and the surrounding retinal tissue [62]. NO has been recognized as an important regulator of ocular blood flow, specifically in the choroid, optic nerve and retina [63,64]. This data suggests that an alternative therapy to lowering the IOP could be using NO directed at treating the optic nerve to manipulate and improve vascular regulation and prevent glaucoma damage [47,63,64]. An example of such an attempt is with NCX 434, a NO-donor triamcinolone acetate compound that has been shown to improve optic nerve head oxygenation and retinal vasculature vasodilation in monkeys [47,60,65].

## 8. NO and AMD

Oxidative stress has been implicated in many chronic disease processes, such as AMD. Reduced NO levels were detected in the plasma of patients with AMD as compared with the control subjects [66]. The impaired availability of NO is also associated with the

enhanced synthesis of endothelin-1 (ET-1), which is a potent vasoconstrictor agent [67]. Consequently, increased ET-1 and decreased NO levels may induce vasoconstriction in small-caliber blood vessels and, thus, be partly associated with choriocapillary ischemia in AMD patients. This was supported in a study that reported elevated levels of ET-1 in patients with AMD as compared to the control subjects [68]. Yuksel at al. suggested that the AMD reduction of endothelial function through reciprocal regulation of the NO and ET-1 levels may explain the decreased choroidal blood flow and, to some extent, the development of choroidal neovascularization in AMD [69].

Another study suggested NO-releasing molecules to protect against oxidative stress on retinal pigmented epithelial cells [70].

## 9. NO and Retinopathy of Prematurity (ROP)

Early reports have shown that mutant alleles of endothelial nitric oxide can lead to a significant reduction of the nitric oxide levels by decreasing the enzyme activity [71,72]. Since endothelial nitric oxide affects the vasculature and the retinopathy of prematurity (ROP) is a vascular disease, one study examined the association of genotypes with the ROP. The results suggested a need for endothelial nitric oxide single-nucleotide polymorphism (SNP) testing on those premature infants with ROP to evaluate their endogenous capacity of NO production [73].

Another aspect of NO and ROP is a link to resveratrol. Resveratrol is a common phytoalexin that has been detected in more than 70 plant species and a few edible materials, such as grape skins, peanuts and red wine. It responds to stress, injury, ultraviolet irradiation and fungal infection [74]. Resveratrol influences retinal injuries by decreasing the oxidation and proliferation of human retinal pigment epithelial (RPE) cells via extracellular signal-regulated kinase inhibition [74].

One study found that resveratrol exerts protective effects via the modulation of NO-mediated mechanisms in both in vivo and in vitro oxygen-induced retinopathy models [75].

## 10. NO and Myopia

Research has shown that exposure to outdoor light, which affects light receptors and the natural circadian rhythm, is the best natural way to stimulate dopamine [76,77]. There is a lot of data supporting the hypothesis that dopamine is one of the neurotransmitters in the retina signaling eye growth in infancy and childhood.

Evidence suggests that dopamine stimulates the synthesis and release of NO. NO operates with the enzyme group NOS to regulate eye growth [78,79].

There is a close interaction between the muscarinergic and dopaminergic systems, as well as between the dopaminergic pathways and NO. The vast distribution of dopaminergic amacrine cells in the retina, as well as the discovery that changes in dopamine levels can be locally induced by local retinal deprivation, are in accordance with the hypothesis that dopaminergic mechanisms control both central and peripheral eye growth [79,80].

Like dopamine, NO is known to mediate light-adaptive changes in the retina, and its synthesis and release are increased by intense or intermittent light. It has been reported that NOS inhibitors block the prevention of experimentally induced form-deprived myopia [79]. Atropine and NO sources inhibit form-deprived myopia in a dose-dependent manner, and NOS inhibitors block the atropine-mediated inhibition of myopia progression. Intraocular NO inhibits myopia dose-dependently, and its production is required for the inhibition of myopia by atropine [78,79]. More research is required to understand the exact mechanisms by which atropine helps prevent myopia, but its dependence on NO is a key point, prompting possible new treatment options. If a treatment were to target NO instead of the muscarinic acetylcholine receptor (mAChR) mechanisms, it would allow control over eye growth without the side effects of photophobia, glare and loss of accommodation [78,81–83]. Light therapy may be another option, which would not require too much specialized equipment or expose the patient to pharmaceuticals and their potential side

effects. To target NO as the therapeutic mediator could lead to a fundamental shift in the treatment of myopia [78].

## 11. Conclusions

The roles of NOS enzyme isoforms and their free radical NO in ocular homeostasis and pathology—specifically, glaucoma and retinal ischemia-associated diseases continue to surface. Scientists persist to attempt to create treatments targeting the NO network, trying to control the NO challenges of a short duration of efficacy, low corneal tissue penetration and inadequate concentration when used in a topical vehicle. The complex heterogeneity of diseases and inflammation further complicates the efficacy of such a targeted therapy. Ultimately, as data emerges, it will help to treat specific cases more efficiently. The ophthalmic community continues to utilize this information as well from the cardiovascular and systemic literature surrounding NO and its isoform synthetase enzymes to help combat disease.

**Author Contributions:** Conceptualization, H.O. and N.E.; methodology, H.O., N.E. and N.L. (Naomi London); data curation, H.O., N.E. and N.L. (Naomi London); writing—original draft preparation, N.E. and N.L. (Nadav Levinge); writing—review and editing, H.O. and N.L. (Nadav Levinge). All authors have read and agreed to the published version of the manuscript.

**Funding:** This research received no external funding.

**Institutional Review Board Statement:** Not Applicable.

**Informed Consent Statement:** Not Applicable.

**Data Availability Statement:** Not Applicable.

**Conflicts of Interest:** The authors declare no conflict of interest.

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
