# Peer review of "Nitric Oxide Interaction with the Eye"

_2411-5150, 2021_

Round 1

Reviewer 1 Report

This manuscript by Erdinest et al reviewed the current progress of NO interaction with the eye, and the role of NO in treatment of a variety of eye disease conditions. The manuscript is clearly written and very informative, the references citation is appropriate, I only have some minor suggestions for this manuscript, and it is suitable for publication after minor revision.

  1. From line 70, there is too much space between “ rior” and “segments”;
  2. From line 94, you may delete “in”;
  3. From line 106, please delete “-“;
  4. The reference cited numbers should be “1.” “2.” “3.”, etc., not “.1” “.2” “.3”, etc.;
  5. From line 227, please delete “â–„”;
  6. There are several references with missing journal names (or book names) or missing page numbers, such as refs. 45, 49, 51,53, 55, etc.;
  7. From page 320, please delete “-“.

Reviewer 2 Report

This paper is a well written mini review regarding the role of nitric oxide (NO) in the eye.  It is appropriate for the journal and interesting to readers in the field.  However, NO is a free radical that reacts with superoxide anion to form peroxynitrite which affects oxidative phosphorylation.  A separate section describing NO and oxidative stress, and its interaction with superoxide anion would add to the paper.  Also, the role of NO in the developing eye, including diseases such as retinopathy of prematurity, will also be important.
